# Impact of the Physical Cellular Microenvironment on the Structure and Function of a Model Hepatocyte Cell Line for Drug Toxicity Applications

**DOI:** 10.3390/cells12192408

**Published:** 2023-10-05

**Authors:** Benjamin Allcock, Wenbin Wei, Kirsty Goncalves, Henry Hoyle, Alisha Robert, Rebecca Quelch-Cliffe, Adam Hayward, Jim Cooper, Stefan Przyborski

**Affiliations:** 1Department of Biosciences, Durham University, Durham DH1 3LE, UK; benpallcock@outlook.com (B.A.); wenbin.wei2@durham.ac.uk (W.W.); kirsty.e.goncalves@durham.ac.uk (K.G.);; 2European Collection of Authenticated Cell Cultures, Salisbury SP4 0JG, UK; 3Reprocell Europe Ltd., Glasgow G20 0XA, UK

**Keywords:** mechanotransduction, bioengineering, 3D cell culture, cytoskeleton, hepatocyte, functionality, cytotoxicity, microenvironment

## Abstract

It is widely recognised that cells respond to their microenvironment, which has implications for cell culture practices. Growth cues provided by 2D cell culture substrates are far removed from native 3D tissue structure in vivo. Geometry is one of many factors that differs between in vitro culture and in vivo cellular environments. Cultured cells are far removed from their native counterparts and lose some of their predictive capability and reliability. In this study, we examine the cellular processes that occur when a cell is cultured on 2D or 3D surfaces for a short period of 8 days prior to its use in functional assays, which we term: “priming”. We follow the process of mechanotransduction from cytoskeletal alterations, to changes to nuclear structure, leading to alterations in gene expression, protein expression and improved functional capabilities. In this study, we utilise HepG2 cells as a hepatocyte model cell line, due to their robustness for drug toxicity screening. Here, we demonstrate enhanced functionality and improved drug toxicity profiles that better reflect the in vivo clinical response. However, findings more broadly reflect in vitro cell culture practises across many areas of cell biology, demonstrating the fundamental impact of mechanotransduction in bioengineering and cell biology.

## 1. Introduction

It is widely recognised that traditional 2D cell culture systems in vitro are less representative of the native physiological microenvironment [1]. Despite this, it remains the primary in vitro approach utilised in many applications. Cells interact with their culture environment both mechanically and biochemically, with mechanotransduction thought to be an important functional driver involved in 2D-associated changes [2]. Mechanotransduction involves cellular interaction with the flat, growth substrate, through junctional and adhesion proteins, leading to cell flattening and cytoskeletal alterations [3]. This in turn impacts nuclear structure and results in the activation and shuttling of messenger proteins from mechanosensitive sites [4]. Ultimately, this change in cell and nuclear structure impacts cell function through changes in gene expression [5,6], reducing the physiological relevance of the system.

The wide-ranging impact of mechanotransduction is particularly important to consider when bioengineering tissues for developing predictive assay systems. For example, the drug development pipeline has a greater than 96% failure rate [7], one contributing factor to which is that 2D culture platforms are too far removed from their native tissue to provide reliable and predictive data when translated into organism or human trials [8].

Modelling liver responses in vitro is often an important step in drug development and safety profiling to determine any toxic effects [9]. A wide range of biological functions are carried out within the liver, and this lends itself to a sizeable body of research devoted to in vitro modelling, aiming to reflect these functions in vitro.

The HepG2 cell line, an immortalised cell line derived from human hepatocellular carcinoma with an unlimited lifespan and stable phenotype [10], is often favoured in liver research. These cells have significant limitations, namely impaired metabolic activity compared to primary hepatocytes. Improvement to biomarker expression has been a desirable outcome of in vitro modelling. At a transcriptomic level, 30% of the HepG2 transcriptome is unique to the cell line [11]. Despite these limitations HepG2 cells are versatile, robust, well characterised and still widely used for drug screenings.

Due to the unique three-dimensional architecture of the liver, there is an increasing recognition of the importance of 3D cell culture as a methodology to recapitulate the physiological structure. Functional differences in HepG2 cells cultured in 2D or 3D have already been reported, with 3D cells demonstrating an artificially enhanced functional profile. Elevated albumin production and CYP450 activity has been observed in simple spheroids compared with traditional 2D monolayer cultures [12]. This raises potential issues with drug sensitivity, as spheroids have demonstrated a decreased response to toxicity [12,13], which may be problematic for drug-induced liver injury screening and contribute to poor predictive outcomes [14]. It has, however, been proposed that 3D culture can offer an improved predictive capability for genotoxicity [15] and may more accurately reflect physiological cancer responses [12,16].

Therefore, there is a need for a standardised and improved approach to better reflect aspects of liver physiology in vitro to provide a better predictive platform for pharmaceutical screenings. This will reduce cost, time and invasive sample collection during clinical trials, whilst improving success rates. Although it is widely accepted that 3D culture offers a more physiologically relevant microenvironment, the extent of that enhancement at the genetic level remains largely unknown.

In this study, we describe the enhancement of HepG2 function for drug toxicity screening through 3D culture and probe underlying changes at both gene and protein levels. We utilise a porous, inert, polystyrene scaffold, Alvetex^®^ (Beltsville, MD, USA), that is routinely used in a wide array of tissue engineering applications [17,18,19,20,21,22,23,24,25,26,27]. This study utilises the mid-pore-size variant, Alvetex^®^ Strata (average 15 µm), which provides 3D growth cues whilst preventing infiltration, promoting a close cell–cell contact reminiscent of native hepatocytes.

We explore 3D culture-mediated changes in cell morphology, leading to differences in cytoskeletal and nuclear architecture, leading to expression changes of basic cellular and liver-specific target genes. We also demonstrate that a novel approach of cell “priming”, by culturing cells for a short period in 3D prior to use, can improve their downstream function.

## 2. Materials and Methods

### 2.1. HepG2 2D Culture

Commercially available HepG2 (ECACC 85011430) cells (UKHSA, Porton Down, UK)—human hepatocellular carcinoma cells originally derived from a 15 year old Caucasian male—were used as a model cell line throughout this study. They were grown in minimal essential media (MEM) containing non-essential amino acids (ThermoFisher Scientific, Loughborough, UK), and supplemented with 2 mM glutamine (ThermoFisher Scientific) and 10% Foetal Bovine Serum (FBS, ThermoFisher Scientific). Cultures were maintained in culture flasks (Greiner Bio-One, Kremsmünster, Austria) at 37 °C, 5% CO_2_ in a humidified environment and passaged at 80% confluence.

### 2.2. HepG2 3D Culture

Alvetex^®^ Strata (Reprocell Europe Ltd., Glasgow, UK), an inert, porous, polystyrene scaffold, was used in this study, with a Young’s modulus of 77 kPa; however, the elasticity of the substrate is thought to have little impact on cellular dynamics due to its porosity (>90%) [28]. Inserts were pre-treated by immersion in 70% ethanol to render them hydrophilic prior to cell seeding. Growth medium was added until it reached the bottom of the Alvetex^®^ insert and 500 μL or 250 μL of growth medium were then added to the top of the membrane of 6-well or 12-well inserts, respectively. Cells were seeded onto the surface of the scaffold at a density of 1 × 10^6^ cells per 6-well insert, or 5 × 10^5^ cells per 12-well insert. As is commonplace in 3D cell culture practices, due to the increased surface area of the scaffold, cells were seeded at high density to promote cell layering and stratification on top of the 3D growth material. Culture medium was changed every 2 days and after 8 days in culture, either harvested for analysis or cells liberated for downstream analysis.

To liberate cells from 3D culture, scaffolds were washed in sterile PBS and then incubated in 0.25% trypsin (ThermoFisher Scientific) for 15 min at 37 °C, 5% CO_2_. Following incubation, soft cell scrapers (Sarstedt, Nümbrecht, Germany) were used to gently scrape the membranes, ensuring that the cell layer was fully removed. Two-dimensional-primed counterparts were seeded at a density of 5 × 10^5^ per well of a 6-well culture plate, cultured for 8 days and liberated using this method. This suspension was collected, trypsin neutralised and centrifuged at 1000 rpm for 3 min.

### 2.3. Spheroid Culture

HepG2 cells were re-seeded into hanging drops after 2D or 3D priming. This was performed using a technique adapted from [15]. HepG2 cells were seeded at a density of 1 × 10^3^ cells per 20 μL drop of growth medium on the inside of the lid of a humidified 9.6 cm Petri dish (ThermoFisher Scientific). Another 20 μL of media was added on day 4 of growth, and cultures were maintained for 7–10 days at 37 °C, 5% CO_2_, in a humidified environment.

### 2.4. Toxicity Testing

Stock solutions containing different xenobiotic compounds (Gemfibrozil (Sigma Aldrich, G9518), Isoniazid (Fisher Scientific, 10225120), Amiodarone (Fisher Scientific, 15502233), Ibuprofen (Sigma Aldrich, I4883), Tamoxifen (Fisher Scientific, 11445161) and Methotrexate (Sigma Aldrich, M9929)) were made using dimethyl sulfoxide (DMSO), EtOH or culture medium as a vehicle. Drug concentrations were based on prior literature, or on 100x the C_max_ reported in human plasma [29,30]. When dissolved in media, vehicle concentrations did not exceed 0.5% and vehicle controls were accounted for.

Appropriate dilutions of compounds were added to the culture medium of both 2D and 3D cultures at day 7 and hanging drop cultures at day 6 (as depicted in Figure 1). Cultures were incubated with compounds for 24 h and media were collected for immediate LDH quantification.

### 2.5. Human Tissue

Fixed human liver tissue was collected by Biopta (Glasgow, UK) under appropriate ethical protocols in compliance with local laws and regulations. The liver sample acquired was from a middle-aged healthy female donor.

### 2.6. Neutral Red Staining

To visualise the gross morphology of cell populations, neutral red was used to stain both 2D and 3D cultures. To stain, 500 μL of neutral red solution (Sigma Aldrich) was applied to unclipped 3D scaffolds and 2D cultures. Plates were left on an orbital shaker for 10 min and wells were washed 5 times with PBS. Staining was visualised via brightfield microscopy.

### 2.7. Scanning Electron Microscopy

Cells were either primed on 2D or 3D growth substrates for 8 days and then liberated and grown on 2D silicon chips for 1 day prior to visualisation.

Cells were fixed in Karnovsky fixative (Sigma Aldrich) for 10 min before washing for 5 min in a 0.1 M phosphate buffer. Samples were treated with 1% Osmium tetroxide diluted in 0.1 M phosphate buffer for 10 min followed by dehydration through a series of ethanol washes. Samples were then dried using a Bal-tec CPD 030 critical point dryer and were then platinum-coated using a Cressington 328UHR and imaged on the Hitachi S-5200 Scanning Microscope at 15 kV.

### 2.8. Immunofluorescence and Fluorescent Staining

Three-dimensional samples were processed as previously described [21], and coverslips, upon which cells were adhered, were blocked and permeabilised with a solution of 0.4% Triton X-100 and 20% NGS for 1 h at room temperature. Primary antibodies (Appendix A) were diluted in blocking buffer and added to samples overnight at 4 °C. Samples were subsequently washed 3 times in PBS for 10 min, before incubation with the secondary antibody diluted in PBS for an hour at room temperature (donkey anti-rabbit Alexa^®^ Fluor 488 or donkey anti-mouse Alexa^®^ Fluor 594, ThermoFisher Scientific, 1:1000). Finally, slides were washed three times in PBS for 10 min and nuclei counterstained with Hoechst-33342 (ThermoFisher Scientific, H3570, 1:10,000) in PBS for 30 s before mounting in VECTASHIELD^®^ (Vector Laboratories, Peterborough, UK).

To visualise the F-actin cytoskeleton, a phalloidin stain (acti-stain 488 phalloidin, Cytoskeleton, Inc., Denver, CO, USA, PHDG1) was added in the final PBS washes after the secondary antibody incubation for 30 min, and the final washing and mounting steps followed as described above.

### 2.9. Light Microscopy

Histology images were captured using Leica ICC50 high-definition camera and Brightfield microscope. Immunofluorescence images were taken using the Zeiss 880 confocal microscope with Airyscan (latest version 2) and Zen (latest version v 13.0) software.

### 2.10. Western Blotting

Cells were lysed using Mammalian Protein Extraction Reagent (M-PER™, ThermoFisher Scientific), supplemented with 1× Halt™ Protease and Phosphatase Inhibitor Cocktail (ThermoFisher Scientific).

Samples were diluted 3:1 with 4× Laemmli sample buffer (Bio-Rad, Hertfordshire, UK) and in 10% 2-mercaptoethanol (Sigma-Aldrich, St. Louis, MO, USA) at 95 °C for 5 min to denature proteins. Samples were then electrophoresed on a polyacrylamide gel at 120 V for up to 90 min in a buffer consisting of 3.028 g Tris, 14.41 g Glycine and 1 g sodium dodecyl sulphate (SDS) in 1 L dH_2_O. The gel was then transferred onto a nitrocellulose membrane (GE Healthcare, Buckinghamshire, UK) in a buffer consisting of: 3.03 g tris, 14.41 g glycine, 200 mL methanol, 800 mL H_2_O, for 20 h, 15 V at 4 °C.

Membranes were blocked for 1 h at room temperature in 5% (*w*/*v*) dried milk diluted in blot rinse buffer (BRB, 1.21 g Tris, 8.8 g NaCl, 0.327 g EDTA, 1 mL Tween20, in 1 L dH_2_O). Membranes were then incubated with a primary antibody (Appendix A) overnight at 4 °C and were then washed in BRB. Secondary antibody was added for 1 h at room temperature (anti-mouse-HRP, Sigma-Aldrich, A4416, 1:5000), followed by subsequent washes in BRB. Finally, Clarity™ ECL detection kit (Bio-Rad) was applied for 5 min, before exposure with photographic film (ThermoFisher Scientific).

### 2.11. Analysis of Cell Culture Medium

Concentration of albumin in the culture medium was determined using a commercial enzyme-linked immunosorbent assay (ELISA) (AssayPro, Saint Charles, MO, USA, EA3201-1) as per manufacturer’s instructions. The assay was read on a BioTek™ ELx800 (Winooski, VT, USA) plate reader at a wavelength of 450 nm. Albumin concentration normalised to total protein concentration as determined using a commercially available Bradford assay kit (Bio-Rad, Hercules, CA, USA).

Urea excretion was measured through a commercial QuantiChrom™ Urea Assay Kit (BioAssay Systems, Hayward, CA, USA, DIUR-100), and manufacturer’s instructions were followed. The assay was read at 520 nm on a BioTek™ Synergy™ H4 Hybrid Multi-Mode Microplate Reader.

### 2.12. Cytotoxicity Assay

For testing the toxicity of xenobiotic compounds, lactase dehydrogenase (LDH) secretion was measured and the commercial Pierce™ LDH Cytotoxicity Assay Kit (Thermo Fisher Scientific) was used as per manufacturer’s instructions. Absorbance was measured at 490 nm and 680 nm using a BioTek™ Synergy™ H4 Hybrid Multi-Mode Microplate Reader and percentage cytotoxicity was calculated.

### 2.13. RNA Extraction and Quality Control

Human liver tissue was snap-frozen in LN_2_ prior to RNA processing. The tissue was dissected into small samples of less than 20 mg and homogenised.

Both homogenised tissue and liberated cells were lysed using: Qiagen RNeasy^®^ mini plus (Qiagen, Hilden, Germany), and by following manufacturer’s instructions. RNA was quantified using a Nanodrop Spectrophotometer ND-100™.

For quality control (QC) purposes, RNA integrity was quantified through the use of an Agilent TapeStation (Agilent, Santa Clara, CA, USA). This provided a RIN score, which, if 8 or greater, was accepted for sequencing.

### 2.14. RNA Sequencing

RNA libraries were assembled within the Department of Biosciences at Durham University. NEBNext Poly(A) mRNA Magnetic Isolation Module and NEBNext Ultra II Directional Library Prep Kit for Illumina (New England Biolabs, Ipswich, MA, USA) were used to create the libraries.

Samples were sequenced using 125 bp paired-end sequencing on HiSeq 2500 using v4 chemistry, providing 29 to 35 millions of read pairs per sample. The adaptors used were NEBNext^®^ Multiplex Oligos for Illumina^®^ and the raw data were returned as FASTQ files. Metadata of RNA-seq data from 1-day cultured 2D primary hepatocytes were obtained from ArrayExpress (all information relating to cell culture and treatments can be viewed at: https://www.ebi.ac.uk/biostudies/arrayexpress/studies/E-MTAB-5984 (accessed on 1 February 2022)) and the corresponding raw data of samples ERS1875912, ERS1875913 and ERS1875914 in FASTQ format were downloaded from ENA (https://www.ebi.ac.uk/ena/browser/home (accessed on 1 February 2022)). The raw data were processed using Salmon [31] (v0.14.1) with human transcriptome Homo_sapiens.GRCh38.cdna.all.fa.gz of 6nsemble release 98 to obtain transcript abundance, which was summarized into read count per gene using tximport [32] (v1.12.3). Genes with read counts of ≥10 in 2 or more samples were kept. Differentially expressed genes were identified using DESeq2 [33] (v1.24.0) with the criteria of adjusted *p* value less than 0.05. Gene expression heatmaps were produced using the heatmap.2 function from gplots package (https://github.com/cran/gplots (accessed on 1 February 2022)). Gene set enrichment analysis of biological processes and pathways was performed using WebGestalt [30].

### 2.15. RTqPCR

RNA isolation followed the same method as described for RNAseq; then, RNA samples were converted to cDNA using the high-capacity cDNA reverse transcription kit (ThermoFisher Scientific) as per manufacturer’s guidelines. A total amount of 1 μg of cDNA was propagated using a Biometra T1 thermal cycler thermocycler with the following parameters: 25 °C for 10 min, 37 °C for 120 min, 85 °C for 5 min and 4 °C to finish. Samples were then diluted with nuclease-free water to make a final concentration of 10 ng μL^−1^.

Predesigned KiCqStart^®^ SYBR^®^ Green Primers (Sigma-Aldrich) were used (Appendix A).

Data were analysed through normalisation to the three reference genes (*HPRT1, TOP1, UBC*) using the ΔΔCt method [34] and averaging the fold change after this normalisation.

### 2.16. Cell Area Quantification

Cell area was quantified using Image J software v. 1.53o. The scale was set in the software and freehand line tool was used to trace the outline of individual cells from high-resolution phalloidin-stained microscopy images.

### 2.17. Statistical Analysis

RNAseq data were analysed as previously described in the RNA Sequencing section. All other data were analysed in GraphPad Prism software (latest version v 10.0.3.). Statistical significance was measured either using a paired/unpaired two-tailed *t*-test or two-way ANOVA with Tukey’s post hoc, as appropriate. * = *p* ≤ 0.05, ** = *p* ≤ 0.01, *** = *p* ≤ 0.001, **** = *p* ≤ 0.0001

## 3. Results

An original pool of HepG2 cells was propagated in 2D prior to priming on either 2D or 3D growth matrices (Figure 1). Cells were then either seeded onto 2D tissue culture-treated polystyrene or onto a 3D polystyrene, porous scaffold. Cells were primed on each substrate for 8 days prior to enzymatic liberation.

### 3.1. Enhanced Cellular Morphology following 3D Culture

HepG2 cells were primed either on 2D or 3D growth substrates for 8 days prior to liberation and seeding on 2D glass coverslips to allow for microscopic analysis (Figure 2(Aa)). Neutral red staining highlights a gross appearance at a population level, with 2D-primed cells appearing as a conventional monolayer (Figure 2(Ab,c,f)), whereas 3D-primed cells (Figure 2(Ad,e,g)) formed distinct colonies, suggesting an increased role of cell-to-cell interactions.

The visualisation of the actin cytoskeleton (green) revealed clear morphological changes in 2D (Figure 2(Ah)) and 3D-primed populations of HepG2 cells (Figure 2(Ai)). The 2D-primed cells had a larger area (Figure 2(Al)), whereas the 3D-primed cells appeared rounder, with less cell spreading. Ultrastructural analysis via scanning electron microscopy (SEM) revealed that 3D-primed cells (Figure 2(Ak)) were more spherical than their 2D counterparts (Figure 2(Aj)), with a greater incidence of microvilli and cytoskeleton protrusions.

Characterisation was also conducted on 2D/3D-primed cells seeded into a 3D culture environment following liberation from their priming substrate (Figure 2(Ba)). Neutral red staining highlights the population-level morphology. The 3D-primed cells clearly formed distinct colonies (Figure 2(Ac,e)), whereas the 2D-primed cells formed a more consistent and homogenous layer across the scaffold (Figure 2(Bb,d)).

### 3.2. Cellular Morphology Impacts Nuclear Structure in 3D Culture

To elucidate the impact of cytoskeletal alterations on nuclear structure, we primed HepG2 cells on 2D/3D growth substrates for 8 days prior to harvest for the protein-level analysis of nuclear structural components (Figure 3A).

Western blot analysis was carried out on the nuclear structural markers SUN1 and SUN2: components of the linker of nucleoskeleton and cytoskeleton (LINC) complex (Figure 3(Ba)). The expressions of both SUN1 and SUN2 were notably increased in 2D-primed cells and confirmed to be statistically significant through densitometry (Figure 3(Bb,c)).

These data were supported via immunofluorescence analysis of SUN1 (Figure 3(Ca,b)) and SUN2 (Figure 3c,d). A more intense staining of both SUN1/2 was apparent in 2D-primed populations compared with 3D. SUN1 appeared localised to the nucleus in 3D-primed cells as opposed to cytoplasmic in the 2D-primed population. This may suggest binding site saturation at the nuclear envelope [35]. This suggests that changes in cellular morphology are transduced through to the LINC complex and 2D-cultured cells develop stiffer nuclei due to increased SUN expression and linkage with the actin cytoskeleton.

### 3.3. Changes in Nuclear Structure Lead to Gene Expression-Level Changes in 3D Culture Consistent with Enhanced Hepatocyte Function

HepG2 cells were cultured on 2D/3D growth substrates for 8 days prior to harvesting for RNAseq (Figure 4A). The expression of mechanotransduction-related genes (Figure 4B) and liver-related genes (Figure 4C) were compared among 2D/3D-primed populations, human liver and the published transcriptome of 1-day cultured 2D primary hepatocytes [36].

Eighteen mechanotransduction-related genes were differentially expressed in 3D and 2D-primed populations (DESeq2 adjusted *p* < 0.05). Genes involved in actin organisation, polymerisation and dynamics (actinin-1 (*ACTN1*), actinin-4 (*ACTN4)*, vinculin (*VCL*), *VASP*, zyxin (*ZYX*), *ROCK1*, *ROCK2*, palladin (*PALLD*)) were all upregulated in 2D-primed cells. Additionally, the nuclear lamins (*LMNA*) and sun (*SUN1*, *SUN2*) genes were upregulated in 2D populations. Transcriptional regulators *YAP (YAP1)*, *TAZ (WWTR1)* involved in mechanotransduction were upregulated in 2D-primed cells by approximately 1.5- and 2.2-fold, respectively. This expression level mirrored primary human hepatocytes, whereas the 3D-primed population demonstrated expression that more closely resembled human liver.

Seventeen liver-related genes were differentially expressed in 3D/2D-primed populations (DESeq2 adjusted *p* < 0.05) with a trend of upregulation in the 3D-primed cells. The common hepatic biomarker albumin (*ALB*) increased in expression by 1.69-fold in 3D-primed cells. Fibrinogen gene (*FGA*, *FGB*, *FGG*) expression was also higher in the 3D-primed population compared with 2D. The expression of antithrombin (*SERPINC1*)—a liver-synthesised coagulation enzyme—is also enhanced in 3D-primed cells, as are *SERPINA6*, thrombopoietin (*THPO*) and transthyretin (*TTR*) and protein S (*PROS1*). Hepatocyte nuclear factor 4 alpha and gamma (*HNF4A*, *HNF4G*), both implicated in liver differentiation, are also upregulated in 3D.

qRT-PCR was used as a method of validation to confirm the RNAseq profiles described above. A set of mechanotransduction-associated and functional genes were selected from the RNAseq data and performed as qRT-PCR analysis (Figure 4D) and expressed as the fold change of 3D- compared to 2D-primed populations. Structural genes involved in the process of mechanotransduction were similarly reduced in 3D-primed populations and functional liver-related gene expression was increased. These data supported the RNAseq findings, ensuring the data presented are reproducible across several analytical techniques.

Transcriptomic datasets (Figure 5A) were further analysed using WebGestalt [37] to identify enriched biological processes (Figure 5B) and pathways (Figure 5C). These data showed a substantial pattern of metabolic enrichment in the 3D-primed cells, including steroid, fatty acid and amino acid metabolism, specific to liver functions. Additionally, the enrichment of mitochondrial gene expression was tested, indicative of enhanced hepatic capacity, whereas mechanotransduction-linked genes, including response to mechanical stimulus and cell motility, were upregulated in the 2D-primed population.

Pathway enrichment analysis (Figure 5C) revealed recurring themes in the genes upregulated in 3D-primed cells. Cholesterol metabolism and biosynthesis were strongly enriched (normalised enrichment scores between 2.5 and 3.0) in phase I and II biotransformation and fatty acid biosynthesis, respectively. The coagulation cascade was also enriched by many factors, which are synthesised hepatically. Focal adhesions were enriched in 2D-primed populations, indicative of structural alterations.

### 3.4. Functional Enhancement in 3D-Cultured Cell Populations

Protein analysis was performed to probe for hepatic capability on 2D/3D-cultured HepG2 cells (Figure 6A). Protein expression of albumin and fibrinogen α-chain were measured via Western blotting (Figure 6B), both of which substantially increased in expression in 3D-primed cells. Densitometry confirmed statistical significance in the case of both fibrinogen α-chain (Figure 6C) and albumin (Figure 6D). Production and secretion into the culture medium of both albumin (Figure 6E) and urea (Figure 6F) were measured via ELISA, and urea was significantly enhanced in 3D-primed cultures.

HepG2 cells were primed on 2D/3D surfaces for 8 days prior to the addition of drugs to the culture medium for 24 h before cytotoxicity testing via LDH assay (Figure 7A), from these data, dose–response relationships could be ascertained. Six drugs were selected based on their known toxicity to hepatocytes: Amiodarone (Figure 7B), Gemfibrozil (Figure 7C), Tamoxifen (Figure 7D) and Ibuprofen (Figure 7E), Methotrexate (Figure 7F) and Isoniazid (Figure 7G).

2D-primed cells were consistently more susceptible to drug-induced toxicity than their 3D-primed counterpart. Although the amplitude of this effect varied across the range of drugs tested, it was a consistent finding. Particularly, Amiodarone, Tamoxifen and Gemfibrozil were observed to have increased toxic activity in 2D-primed cultures as opposed to 3D. Isoniazid even displayed cytotoxic activity in 2D, but this was unidentifiable in 3D-primed populations at the concentration range tested in this study.

In order to test the benefit of “priming” for downstream applications, we cultured HepG2 cells for 8 days on 2D/3D substrates, before seeding them into a secondary culture system: hanging drop spheroids (Figure 8A). Spheroids were either harvested for analysis or used for drug toxicity studies, whereby drugs were added to the hanging drop culture medium for 24 h prior to LDH cytotoxicity assessment.

Fluorescent microscopy highlighting the actin cytoskeleton (green), propidium iodide as a marker of cell death (red) and Hoechst-stained nuclei (blue) all reveal the structure of spheroids formed from 2D- and 3D-primed cells (Figure 8B).

Albumin (Figure 8C) and urea (Figure 8D) secretion into the culture medium was significantly enhanced in spheroid cultures formed from 3D-primed cells, as it was in the cultures themselves, providing consistent evidence that functionality is improved in 3D culture.

Drug toxicity to a range of drugs tested during the priming step was also documented for spheroids formed from 2D/3D-primed cells, including: Amiodarone (Figure 8E), Gemfibrozil (Figure 8F), Tamoxifen (Figure 8G) and Ibuprofen (Figure 8H). Although decreased sensitivity to drug-mediated toxicity was still evident for some compounds tested (Gemfibrozil and Tamoxifen), there did appear to be some attenuation of this response, as profiles of drugs such as Amiodarone were indistinguishable between 2D- and 3D-primed populations.

## 4. Discussion

The notion of “priming”—that the microenvironment with which a cell is exposed even for a short period of time can have downstream functional consequences—has been explored throughout the literature [38,39,40]. However, there remains a lack of understanding of the specific effect of the biophysical properties of growth on the structure and functionality of cells. In this study, we have demonstrated the impact of 3D culture at a cell, protein and genomic levels, leading to improvements in cell functionality, physiological relevance and predictive potential. We demonstrate these advances using HepG2 as a model cell line, a commonly used screening technique in pharmaceutical research; however, we believe that the conclusions drawn from this study also apply more broadly to cell biology.

The process of mechanotransduction is initiated through the interaction of the cell membrane with the growth substrate and is transduced via the actin cytoskeleton. The differences in the actin cytoskeleton for cells cultured in 2D or 3D was clear. The 2D culture conveyed structural changes to cells, which appeared larger and flattened, whereas 3D-primed cells had a reduced area, appeared more circular and had a greater number of cytoskeleton protrusions. Interestingly, this enhanced morphology that better represents native hepatocyte morphology in vivo is retained once cells have been liberated from the 3D scaffold.

In turn, the substrate-induced changes to the actin cytoskeleton subsequently impacted the nuclear structure through the upregulation and nuclear localisation of SUN1/2. Changes in nuclear structure consequently impacted gene expression. Immunofluorescence localisation suggests that 2D-cultured cells have a more rigid nucleus, whereas the 3D cells have a malleable nucleus consistent with previously reported findings in mouse fibroblasts cultured in hydrogels of varying levels of crosslinking and therefore stiffness [41,42]. Substrate geometries are also known to induce the shuttling of epigenetic factors that alter the nuclear morphology, stiffness and, ultimately, gene expression. This correlates with the observations we have described, including changes to the cytoskeleton, nuclear structure and gene expression. This suggests that 3D culture induces an altered mechanical state, resulting in epigenetic changes that we have demonstrated to persist after liberation, a phenomenon supported by previous studies [43].

Here, we have provided a thorough investigation of the biological changes occurring at a transcriptional level as a result of changing culture geometry, confirming long-held assumptions that 3D culture provides a more physiologically relevant microenvironment for cell culture. Panels of genes were analysed to generate heatmaps of genes heavily implicated in mechanotransduction or liver function. Hepatic genes were often upregulated in 3D-primed HepG2 cells, whereas genes involved in mechanotransduction were upregulated in 2D-primed cells, with the expression profile of 3D-primed, more closely resembling that of human liver. Importantly, this demonstrates that 3D cells exhibited a more physiologically representative expression of both the functional and mechanotransduction genes.

The uniform increase in many of these genes involved in mechanotransduction-related functions may possibly be a direct result of the increased YAP/TAZ expression. YAP/TAZ is a transcription factor that is very closely implicated in mechanotransduction responsible for promoting the transcription of genes involved in extracellular matrix composition, cell–matrix interaction and cytoskeleton integrity [3,44,45,46]. This correlates strongly with the upregulation of YAP/TAZ in the 2D cells. Nuclear stretching as a result of growth on stiff substrates has been demonstrated as a mechanism through which an increased YAP nuclear translocation can occur due to reduced mechanical resistance in the nuclear pores [47].

Gene enrichment revealed an upregulation of genes associated with functional processes of hepatocytes within the 3D-primed population, including cholesterol biosynthesis, fatty acid metabolism and steroid metabolism, suggesting an improved hepatic function. Only one key drug-metabolising cytochrome (CYP1A2) was differentially upregulated in 3D HepG2 cells to a ≥2-fold significance level, although expression levels were still very low. However, other CYPs, with the exception of CYP2S1, CYP24A1 and CYP4F22, were enhanced in 3D-cultured hepatocytes but to a level that was still significantly lower than primary cells or human liver. This suggests that 3D culture can improve cell functionality and phase I metabolism in general, but also highlights how far removed the HepG2 cell line is from native hepatocytes.

Using a separate orthogonal technique of QRT-PCR, we tested for the same expression changes in a selected number of genes. The results corroborated sequencing data and proved that the wide-ranging biological effects of priming as demonstrated via RNAseq were robust and reproducible.

We then investigated protein-level changes to further support the genomic data, with a consistent increase in albumin production by the 3D-primed cell population with transcriptomics. Both albumin and urea productions were significantly enhanced in the 3D priming model and the 3D-primed spheroids over the 2D counterparts, and this provided further evidence that the synthetic properties of the HepG2 cells were enhanced through priming in a 3D microenvironment.

Despite an increase in detected urea, the urea cycle is known to be diminished in HepG2 cells due to ornithine transcarbamylase (OTC) and arginase 1 (ARG1) deficiency [48]. An increase in the expression of urea cycle genes (including *ARG1* and *OTC*) was seen in the transcriptomic analysis, which could partially explain the increase in urea detection; however, the cycle is likely still deficient compared to primary cells.

For further functional validation, cells were exposed to xenobiotic compounds both in in situ within the priming cultures themselves (2D/3D) and within spheroid structures, acting as a downstream assay system. Historically, results of drug toxicity tests are often variable between different studies, with 3D HepG2 demonstrating varied sensitivity depending on the model and the drugs tested [49]. In these experiments, 3D priming models displayed a decreased sensitivity to xenobiotic compounds. There are many potential reasons for this difference in response to the drugs, one of which is that the metabolic profile of the cells has changed, which is supported by the genomic findings. To further identify functional differences between 2D/3D-primed populations in spheroid structures, transcriptomic analysis could be conducted in the future to determine the pathways involved in the attenuated functional enhancement post-3D priming.

An enhanced drug sensitivity was attenuated when 3D-primed cells were cultured as spheroids. Despite this, 3D-primed spheroids still elicited a delayed cytotoxic response to Gemfibrozil with cytotoxicity observed at a lower drug concentration in the 2D-primed spheroids. This suggests that a level of residual difference in the cells is still present depending on their mechanical priming history. The overall more similar reaction to xenobiotics in spheroids could indicate that the metabolic profiles of the cells may have reached a threshold. However, this initial characterization of 3D-primed HepG2 cells was limited to a small number of commonly tested drugs, the response of cells to which is well regarded. In the future, it would be beneficial to extend this testing to a larger number of drugs with differential effects on the liver with a particular focus on liver-activated xenobiotics. The activation of drugs such as cyclophosphamide though CYP-mediated phase I metabolism by 3D-primed cells would provide valuable insight into any enhanced metabolic activity observed.

## 5. Conclusions

We therefore demonstrate in this study, the importance of microenvironment geometry on cell structure and function. We establish a link between culture substrate, cytoskeletal changes as a result of mechanotransduction, the downstream consequences this has on nuclear structure, gene expression and, ultimately, functionality, evidencing each step of the pathway. We utilise HepG2 as a well-characterised cell line to model aspects of pharmaceutical screenings, an application limited by 2D culture methods. We demonstrate enhanced functionality in this cell line, evidencing the need for more physiologically representative culture methods, to recapitulate native functionality in vitro.

Although we discuss the use of 3D cell culture technology to enhance the functional capability of hepatocytes, we believe these data demonstrate a broader trend in the enhancement of cellular functionality in 3D culture. This suggests that the adoption of 3D culture practices could benefit many aspects of cell biology as a whole, including investigations into: stem cell science, tissue bioengineering, neuroscience, drug testing, industrial applications and fundamental insights.

## Figures and Tables

**Figure 1 cells-12-02408-f001:**
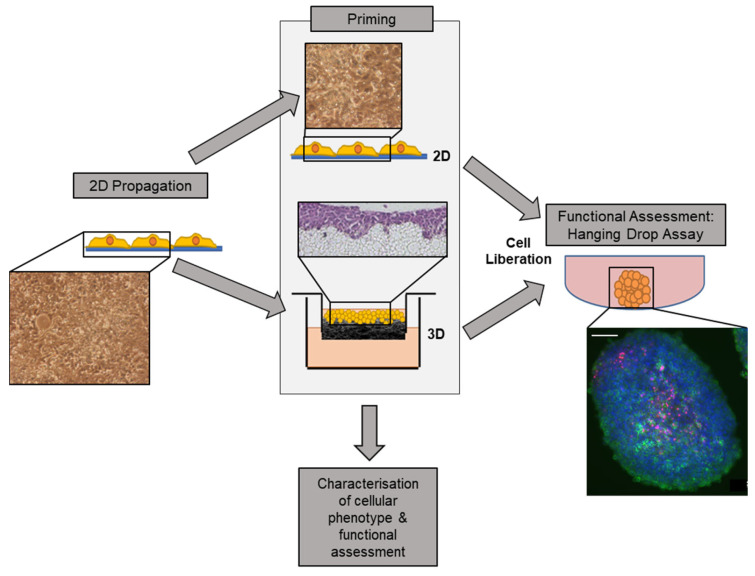
An innovative culture strategy that improves cellular function prior to use in an assay system. HepG2 cells, a well-characterised hepatocarcinoma cell line widely used in drug toxicity testing, was selected as a model cell line. Cells were propagated in traditional 2D culture then seeded onto either a 2D or 3D culture substrate for 8 days in a novel priming step. The 3D culture substrate was an inert porous polystyrene scaffold (Alvetex^®^ Strata), whereas 2D culture substrate was standard culture-ware. Following 8 days of growth on each priming substrate, cells were liberated through trypsinisation and either harvested for analysis and characterisation or re-seeded into a secondary culture system: hanging drop assay, which is a widely used pharmacological assay system.

**Figure 2 cells-12-02408-f002:**
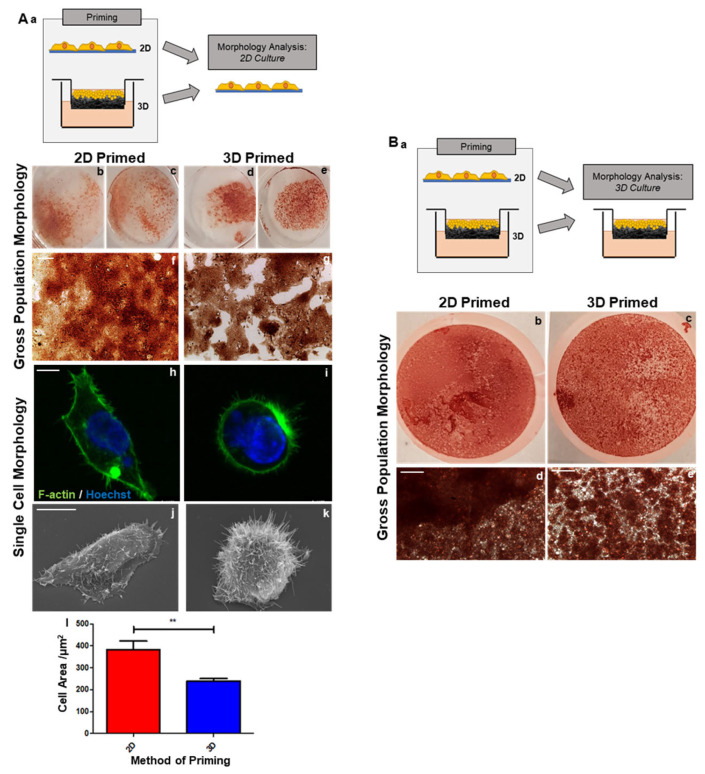
Three-dimensional-primed cells exhibit enhanced morphology at both a cellular and population levels. Schematic representation of HepG2 cells primed on either 2D/3D substrates, liberated and allowed to adhere onto glass coverslips for 24 h prior to morphology observations (**Aa**). Gross population morphology highlighted by neutral red staining (**Ab**–**g**) demonstrates the differential interactions of 2D and 3D-primed cells and their ability to form colonies and close contact with one another. At a single-cell level observed through phalloidin staining of the actin cytoskeleton (green, **Ah**,**i**) and nuclei highlighted in blue, 3D-primed cells appear much more circular with a significantly reduced cell area (**Al**) (data represent mean ± SEM, *n* = 40). Scanning electron microscopy (SEM, **Aj**,**k**) supports this observation and also demonstrates a greater number of surface protrusions in 3D-primed cells. Schematic representation of cells primed on either 2D/3D substrates, liberated and seeded onto 3D culture scaffolds (**Ba**–**c**). Gross population morphology as visualised by neutral red staining (**Ab**–**g**) highlights enhanced colony formation in 3D-primed subpopulations. Scale bars: （**Af**,**g**, **Bd**,**e**） = 200 µm, （**Ah**,**I**） = 5 µm, （**Aj**,**k**） = 10 µm. ** = *p* < 0.01.

**Figure 3 cells-12-02408-f003:**
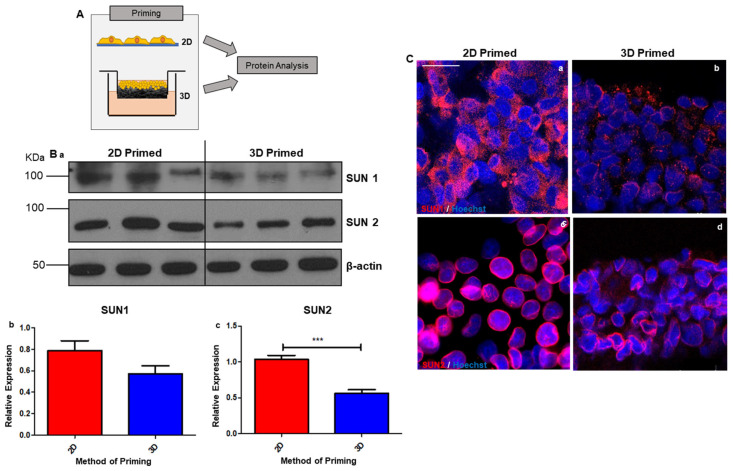
Nuclear structural markers are downregulated in 3D-cultured cells, indicating changes to nuclear stiffness. Schematic representation of HepG2 cells cultured on either 2D or 3D substrates and liberated prior to protein expression analysis (**A**). Expression of nuclear lamina proteins (SUN1 and SUN2) measured via Western blot analysis (**Ba**) reveals marked downregulation in the 3D-primed population. Relative expression of both markers determined by densitometry (**Bb**,**c**) confirms this reduction in the 3D-primed population, normalised to β-actin (data represent mean ± SEM, *n* = 6). Representative immunofluorescence images visualising expression of SUN1 (**Ca**,**b**) and SUN2 (**Cc**,**d**) in 2D and 3D culture cells supports this finding as reduced staining is observable in 3D-cultured samples (SUN1/2 stained red, Hoechst stain highlights nuclei (blue)). Scale bars 20 µm. *** = *p* < 0.001.

**Figure 4 cells-12-02408-f004:**
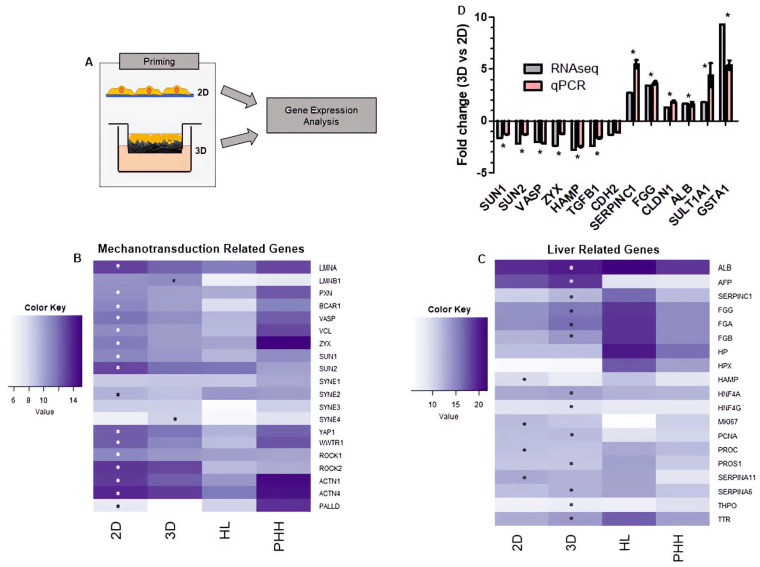
Changes in expression of hepatic and mechanotransduction-related genes in 3D hepatocyte cultures. Schematic representation of HepG2 cells cultured in 2D or 3D conditions and then harvested for gene expression analysis (**A**). Heatmaps of mechanotransduction (**B**,**C**) liver-related gene expression created with Heatmap.2 in R, from average DESeq2-normalised expression values in each. Colour key equates to log_10_ of the expression value; darker colours indicating higher absolute expression (data represent *n* = 4, HL = Human Liver, PHH = Primary Human Hepatocytes). RT-qPCR analysis (**D**) confirms fold change of selected genes from RNAseq dataset, validating findings and reproducibility of dataset (data represent mean ± SEM, *n* = 8). * = *p* ≤ 0.05.

**Figure 5 cells-12-02408-f005:**
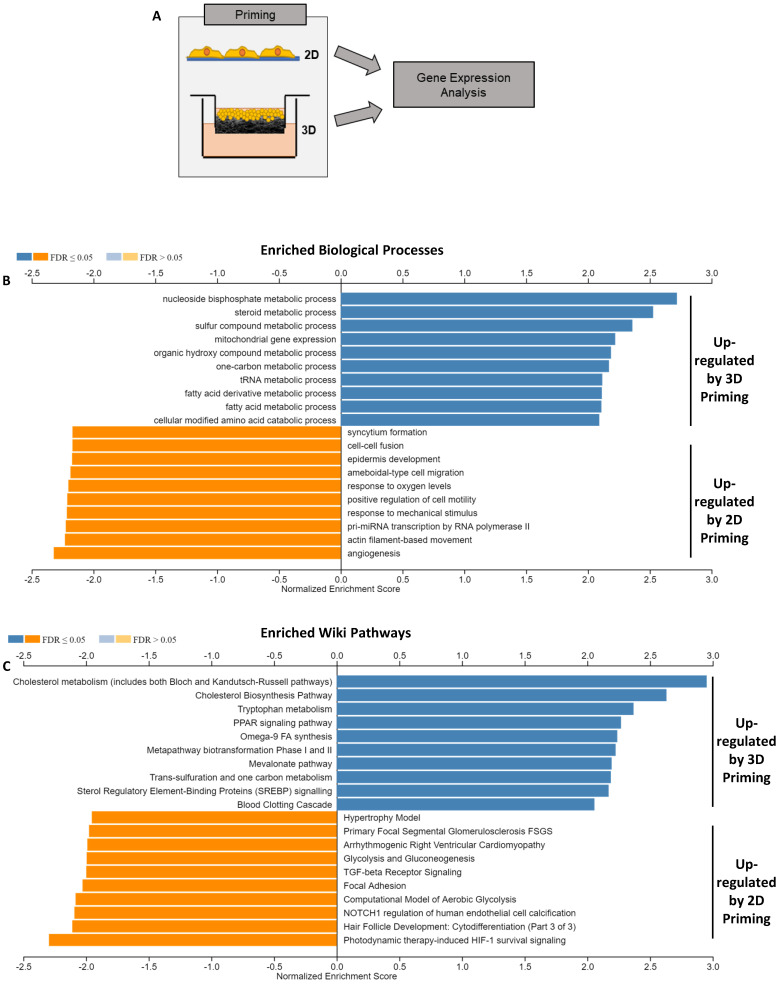
Enhanced metabolism and biosynthesis gene expression in 3D hepatocyte culture through gene enrichment analysis. Schematic representation of HepG2 cells cultured in 2D or 3D conditions and harvested for gene expression analysis (**A**). Gene set enrichment analysis of significantly differentially expressed genes in 2D/3D cultures calculated using log 2-fold change values of the genes (data represent *n* = 4, adjusted *p* value ≤ 0.05). WebGestalt was used to calculate non-redundant enriched biological processes using gene ontology annotation from www.geneontology.org (accessed on 1 February 2022) (**B**), and Wiki pathway data from www.wikipathway.org (accessed on 1 February 2022) (**C**). Length of bars indicates normalised enrichment score with metabolism- and biosynthesis-related genes upregulated in 3D culture, and mechanotransduction-related genes upregulated in 2D culture.

**Figure 6 cells-12-02408-f006:**
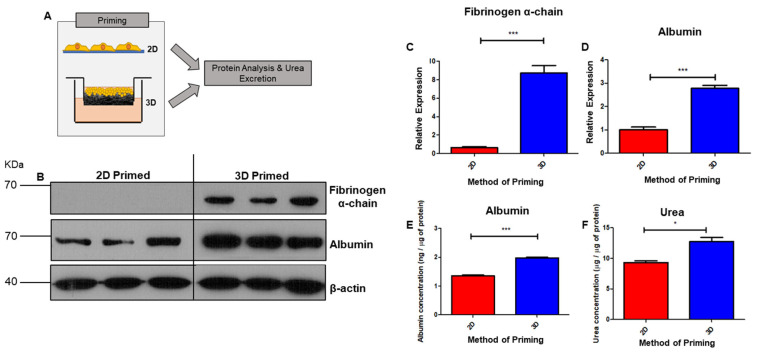
Enhanced protein expression of liver function biomarkers in 3D culture. Schematic representation of HepG2 cells culture in 2D/3D conditions and either harvested for protein analysis or biosynthesis and secretion of organic substances measured through ELISA assay (**A**). Hepatic functional proteins albumin and fibrinogen α-chain are upregulated in 3D-primed cells, as detected via Western blot (**B**) analysis. Densitometry measurements confirm enhancement of relative expression of fibrinogen α-chain (**C**) and albumin (**D**) in 3D cultures, normalised to the loading control β-actin (data represent mean ± SEM, *n* = 6). Increased production of metabolites albumin (**E**) and urea (**F**) in medium of 3D-cultured cells, as measured via ELISA and normalised to total protein (data represent mean ± SEM, *n* = 24). * = *p* ≤ 0.05, *** = *p* < 0.001.

**Figure 7 cells-12-02408-f007:**
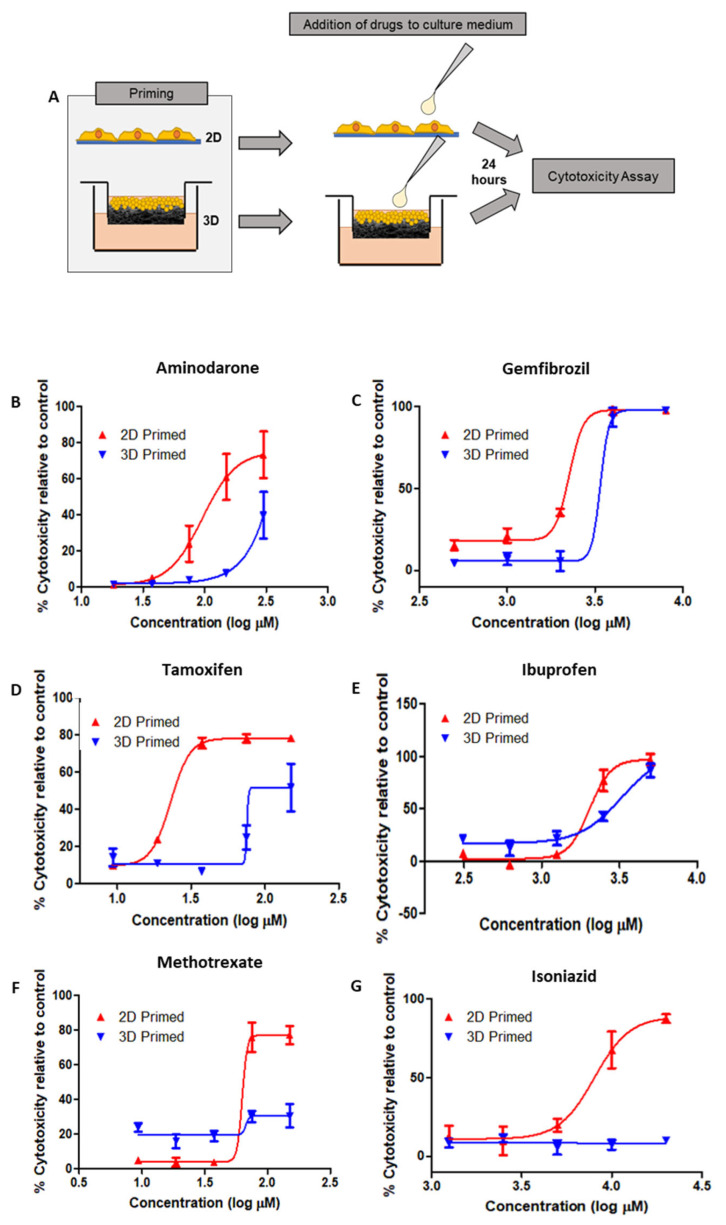
Differential drug toxicity profiles in 3D-primed hepatocytes demonstrate functional-level changes as a result of culture substrate interactions. Schematic representation of HepG2 cells cultured on 2D/3D growth substrates for 8 days before the addition of drugs to the culture medium for 24 h (**A**). Drug toxicity curves were determined from an LDH assay of the culture medium. Cultures were exposed to increasing concentrations of Amiodarone (**B**), Gemfibrozil (**C**), Tamoxifen (**D**), Ibuprofen (**E**), Methotrexate (**F**) and Isoniazid (**G**) (data represent mean ± SEM, *n* = 8). The 3D cultures were generally less susceptible to drug-induced toxicity, suggesting a more reliable cell source for pharmaceutical modelling.

**Figure 8 cells-12-02408-f008:**
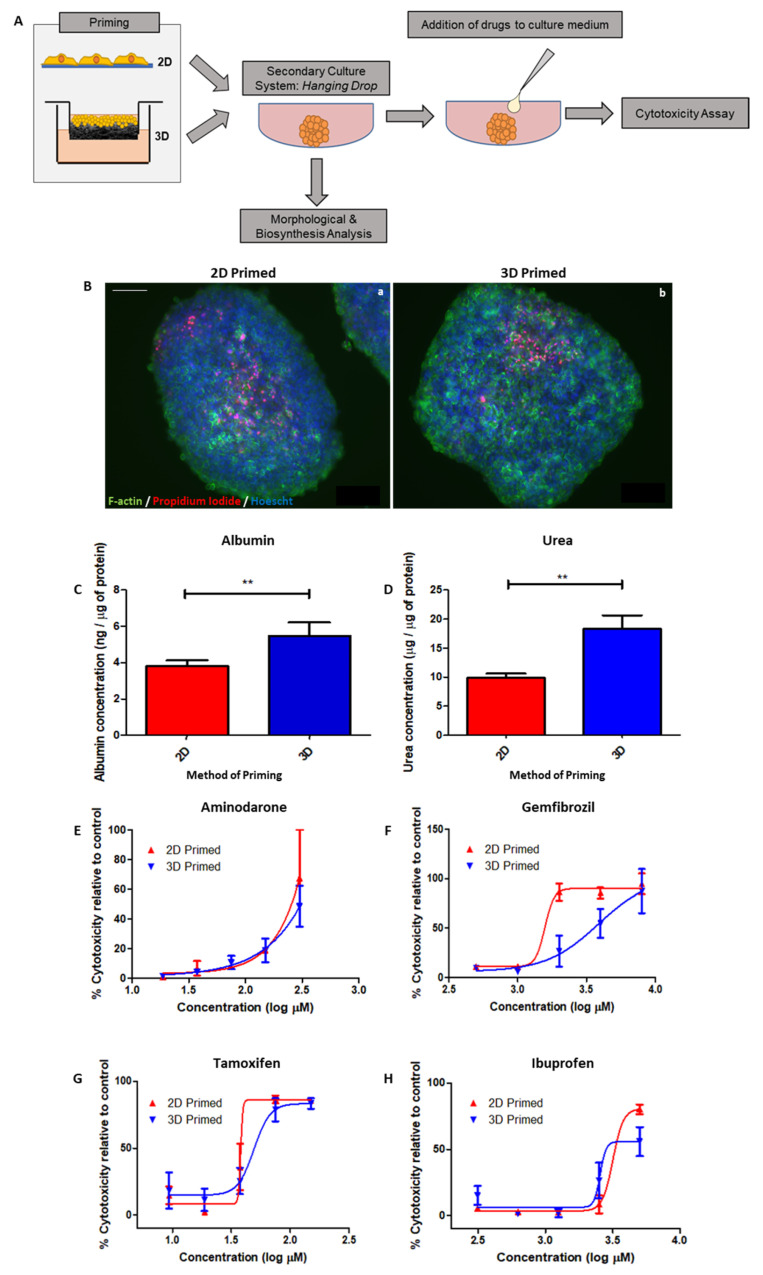
Enhanced functionality of 3D-primed cells in “gold standard” hanging drop assay. Schematic representation of 2D/3D-primed HepG2 cells, re-seeded into a secondary hanging drop culture system to form spheroid structures (**A**). Hanging drops were either analysed or dosed with drugs to develop drug toxicity profiles. Representative images of spheroids formed from 2D (**Ba**)- or 3D (**Bb**)-primed cells. F-actin is stained by phalloidin in green, propidium iodide stains dead cells in red and Hoechst highlights nuclei in blue. Increased production of metabolites, albumin (**C**) and urea (**D**) in the medium of 3D-cultured cells measured via ELISA, remained a consistent finding even when cells were in spheroid form. Measured via ELISA of culture medium and normalised to total protein (data represent mean ± SEM, *n* = 24). Spheroids were exposed to increasing concentrations of Amiodarone (**E**), Gemfibrozil (**F**), Tamoxifen (**G**) and Ibuprofen (**H**) (data represent mean ± SEM, *n* = 8). Drug toxicity curves were determined from an LDH assay of the culture medium. Scale bars: 100 µm. ** = *p* < 0.001.

## Data Availability

The data that support the findings of this study are openly available in GEO National Centre for Biotechnology Information at https://www.ncbi.nlm.nih.gov/geo/query/acc.cgi?acc=GSE236919 (accessed on 1 February 2022), reference GEO accession GSE236919 and token ID: iteduuecxrqtlyr.

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
