# Peer review of "Impact of the Physical Cellular Microenvironment on the Structure and Function of a Model Hepatocyte Cell Line for Drug Toxicity Applications"

_cells, 2023, doi:10.3390/cells12192408_

Round 1

Reviewer 1 Report

The paper of Allcock et al. on the impact of physical cellular microenvironment of hepatocytes adds a significant detail to the many approaches of in vitro toxicology test systems. That is the contribution of mechanotransduction on 2D cultured cells versus cells in a more physiological  3D cultures. Using the widely used HepG2 cell line, they describe that a process called 2D priming significantly increase expression of some genes involved in mechanotransduction while 3D priming seems to signficantly increase expression of certain hepatic functions important for e.g. cholesterol biosynthesis, urea production and even biotransformation. Differences of 2D versus 3D primed HepG2 cells in drug toxicity can be observed for prototype drugs such as Aminodarone, Gemfibrozil, Tamoxifen and Iboprofen. Surprisingly, those drugs seem to be less toxic für 3D primed HepG2 cells.

The paper is well written, interesting and relevant for the field of in vitro toxicology, pharmacology, but also for cell culture technology in general.

I am referring to the data on the toxicity of the drugs. I think they could have used other drugs. Prodrugs whose toxicity is only visible after metabolic activation would be particularly interesting. I am not calling for the authors to perform additional experiments with prototypical liver-activated xenobiotics (e.g., cyclophosphamide, aflatoxin) to prove that 3D-primed hepatocytes perform better than cells from 2D cultures, but I would like to see the authors discuss on this.

In addition, it is well known that conventionally cultured HepG2 cells in 2D have virtually no phase I enzyme expression (with the exception of CYP2B6) and are therefore of very limited use for biotransformation and in vitro toxicology studies. So, if there indeed is upregulation of biotransformation-relevant pathways, I would like the authors to discuss about the relevance of this finding for CYP450 enzymes.

Well written

Minor points:

Introduction,  second para: wrong spelling of „mechanotransduction“

Introduction, page 2 second para: Revise sentence wording „The HepG2 cell line…, is an often favoured in liver research“

Author Response

Reviewer 1

Feedback from Reviewer 1 has been incorporated into the manuscript and is highlighted in green.

Comments and Suggestions for Authors

The paper of Allcock et al. on the impact of physical cellular microenvironment of hepatocytes adds a significant detail to the many approaches of in vitro toxicology test systems. That is the contribution of mechanotransduction on 2D cultured cells versus cells in a more physiological  3D cultures. Using the widely used HepG2 cell line, they describe that a process called 2D priming significantly increase expression of some genes involved in mechanotransduction while 3D priming seems to signficantly increase expression of certain hepatic functions important for e.g. cholesterol biosynthesis, urea production and even biotransformation. Differences of 2D versus 3D primed HepG2 cells in drug toxicity can be observed for prototype drugs such as Aminodarone, Gemfibrozil, Tamoxifen and Iboprofen. Surprisingly, those drugs seem to be less toxic für 3D primed HepG2 cells.

The paper is well written, interesting and relevant for the field of in vitro toxicology, pharmacology, but also for cell culture technology in general.

I am referring to the data on the toxicity of the drugs. I think they could have used other drugs. Prodrugs whose toxicity is only visible after metabolic activation would be particularly interesting. I am not calling for the authors to perform additional experiments with prototypical liver-activated xenobiotics (e.g., cyclophosphamide, aflatoxin) to prove that 3D-primed hepatocytes perform better than cells from 2D cultures, but I would like to see the authors discuss on this.

The following was added to the discussion to encompass this point:

However, this initial characterization of 3D primed HepG2 cells was limited to a small number of commonly tested drugs, the response of cells to which is well regarded. In the future, it would be beneficial to extend this testing to a larger number of drugs with differential effects on the liver with particular focus on liver-activated xenobiotics. Activation of drugs such as cyclophosphamide though CYP-mediated phase I metabolism by 3D primed cells, would provide valuable insight into any enhanced metabolic activity observed.”   Lines 557-563

In addition, it is well known that conventionally cultured HepG2 cells in 2D have virtually no phase I enzyme expression (with the exception of CYP2B6) and are therefore of very limited use for biotransformation and in vitro toxicology studies. So, if there indeed is upregulation of biotransformation-relevant pathways, I would like the authors to discuss about the relevance of this finding for CYP450 enzymes.

The following information regarding CYP expression was added to the discussion:

Only one key drug metabolising cytochrome (CYP1A2) was differentially upregulated in 3D HepG2 cells to a ≥2-fold significance level though expression levels were still very low. However, other CYPs with the exception of CYP2S1, CYP24A1 and CYP4F22 were enhanced in 3D cultured hepatocytes but to a level that was still significantly lower than primary cells or human liver. This suggests that 3D culture can improve cell functionality and Phase I metabolism in general, but also highlights how far removed the HepG2 cell line is from native hepatocytes.”  Lines 519-524

Minor points:

Introduction,  second para: wrong spelling of „mechanotransduction“

This has been corrected. Line 41

Introduction, page 2 second para: Revise sentence wording „The HepG2 cell line…, is an often favoured in liver research“

Removed “an”. Line 50

Reviewer 2 Report

The manuscript by Allcock et al decribes the influence of the physical microenvironment on the structure and functionality of cells for down stream drug toxicity applications. The study utilized HepG2 cells as a cellular model representing hepatocytes, compared the cell performance after a 2D vs 3D pre-cultivating (priming) process and subsequent drug toxicity testing.

The work by Allock et al contributes to the significant field of clarification and characterization after alterations of mechanotransduction in cells.

The statement of the manuscript substantiates the hypothesis of the enhancing effect on morphology and functionality of cultivating cells in a 3D manner, even for a short period.

The manuscript is clearly written and never loses its main thread.

However, the readers of the manuscript could benefit from addressing some minor points:

Materials and Methods

line 93: Could the authors please comment on the cell density seeded? And were the cells also 8 d in culture as the cells seeded on Alvetex?

line 99: the cell density is stated, on the Alvetex website it is shown, that the scaffolds have a different growth areas compared to normal well plate. Please could authors comment on the large amount of cells seeded onto the scaffolds? Just for clarification - is the idea to achieve the 3D layer by the seeding and to supply the cells onto the scaffold from both sides?

Could authors please comment on the E-modulus of Alvetex?

Toxcity testing

Please could authors consider to include Figure 1 here for clarification?

Western Blotting and Immunofluorescence Staining

Did the authors consider to add a table with all used antibodies and clone numbers, suppliers, dilutions etc.?

Could the authors please provide a brief explanation for their choice of specific markers?

line 248 - please add reference

line 277 - please check sentence

Results

Please check Figure 1, as mentioned in the M&M section not only cells in hanging drops were subjected for functional assessment

Please could authors comment on their finding that cells on 2D formed the more confluent layer? Stiffness, gravity ?

line 339 + 340 Please could authors comment on the term primary human hepatocytes? Do authors now how the sample was cultured prior analysis?

Figure 4 D is missing

line 407 -please check albumin is not a metabolite

Figure 7 - please check font in caption and line 429 - cyto- toxicity

Discussion

line 486 - 492 Authors cite previous findings, could authors may address or indicate the cells or methods used from the previous findings? 

line 538  - cells from 3D spheroids show a delayed cytotoxic response - did the authors consider to investigate the activities of prominent Cytochrome P450 enzymes?

Please check the title and consider to include the "hepatocyte model cell line" used in the study

Line 85 and 94- please check, HepG2 culture or model

line 260 - 267 text passage redundant

line 275 - 277 The authors address the morphological changes - it remaines unclear if the authors refer to differences or primary cells.

line 154 - please check Hoechst

Author Response

Reviewer 2

Feedback from Reviewer 2 has been incorporated into the manuscript and is highlighted blue.

Comments and Suggestions for Authors

The manuscript by Allcock et al decribes the influence of the physical microenvironment on the structure and functionality of cells for down stream drug toxicity applications. The study utilized HepG2 cells as a cellular model representing hepatocytes, compared the cell performance after a 2D vs 3D pre-cultivating (priming) process and subsequent drug toxicity testing.

The work by Allock et al contributes to the significant field of clarification and characterization after alterations of mechanotransduction in cells.

The statement of the manuscript substantiates the hypothesis of the enhancing effect on morphology and functionality of cultivating cells in a 3D manner, even for a short period.

The manuscript is clearly written and never loses its main thread.

However, the readers of the manuscript could benefit from addressing some minor points:

Materials and Methods

line 93: Could the authors please comment on the cell density seeded? And were the cells also 8 d in culture as the cells seeded on Alvetex?

Yes, 2D primed cells were cultured for the same length of time as 3D primed cells and this has been rectified in the manuscript - “cultured for 8 days” was added to line 112

line 99: the cell density is stated, on the Alvetex website it is shown, that the scaffolds have a different growth areas compared to normal well plate. Please could authors comment on the large amount of cells seeded onto the scaffolds? Just for clarification - is the idea to achieve the 3D layer by the seeding and to supply the cells onto the scaffold from both sides?

Yes, cell layering was the objective but cells were only seeded onto the top of the scaffold – not both sides. To clarify this in the text, the following was added:

“…onto the surface of the scaffold” Line 102

As is commonplace in 3D cell culture practices, due to the increased surface area of the scaffold, cells were seeded at high density to promote cell layering and stratification on top of the 3D growth material.” Lines 104-106

“…2D primed counterparts were seeded at a density of 5x105 per well of a 6-well culture plate” Line 111-112

Could authors please comment on the E-modulus of Alvetex?

Alvetex® Strata (Reprocell Europe Ltd., Glasgow, UK), an inert, porous, polystyrene scaffold was used in this study, with a Young’s modulus of 77 kPa, however the elasticity of the substrate is thought to have little impact on cellular dynamics due to its porosity (>90%) (Asthana and Kisaalita, 2013).”  Lines 96-99

Toxcity testing

Please could authors consider to include Figure 1 here for clarification?

(as depicted in Figure 1)” was added to line 131

Western Blotting and Immunofluorescence Staining

Did the authors consider to add a table with all used antibodies and clone numbers, suppliers, dilutions etc.?

Information regarding primary antibodies, suppliers, applications, etc. has been included as a supplementary table (see below) and the methods section amended accordingly.

Supplementary Table 2: Primary Antibodies

Primary antibodies and their suppliers, used in both immunofluorescence and western blotting. In both cases primary antibodies were incubated overnight at 4 oC prior to wash steps and addition of secondary antibodies. Rationale supporting their choice and relevance to the study is also detailed above.

Could the authors please provide a brief explanation for their choice of specific markers?

Rationale as to the choice of specific biomarkers has been added to the newly assembled Supplementary Table 1: Primary antibodies (see above).

line 248 - please add reference

This may not have been clear but is referring to early in the methods section, we have added the following to clarify:

“…in RNA Sequencing section.” Line 253

line 277 - please check sentence

Removed the word “are”. Line 290

Results

Please check Figure 1, as mentioned in the M&M section not only cells in hanging drops were subjected for functional assessment

Added “ & functional assessment” to relevant place in Figure 1 schematic

Please could authors comment on their finding that cells on 2D formed the more confluent layer? Stiffness, gravity ?

Replaced the word “confluent” with “consistent and homogenous”. Lines 287-288

line 339 + 340 Please could authors comment on the term primary human hepatocytes? Do authors now how the sample was cultured prior analysis?

As cited in the materials & methods section, the transcriptomic analysis for primary human hepatocytes was downloaded from a data repository. Information regarding their culture environment can be found at the same link. We have added this to direct the reader to the original data should they wish to obtain further information.

Meta data of RNA-seq data from 1-day cultured 2D primary hepatocytes was obtained from ArrayExpress (all information relating to cell culture and treatments can be viewed at: https://www.ebi.ac.uk/biostudies/arrayexpress/studies/E-MTAB-5984) and the corresponding raw data of samples ERS1875912, ERS1875913 and ERS1875914 in FASTQ format was downloaded from ENA (https://www.ebi.ac.uk/ena/browser/home).” Lines 224-225

Culture conditions available via hyperlink state the following details:

“Cryopreserved Primary Human Hepatocytes (PHH, Invitrogen) of 3 individuals (Hu8119, Hu1591 and Hu1540) were thawed for 1 minute at 37C in a water bath. Next, these PHH were pooled in order to bypass inter-individual variability in susceptibility to toxicants and cultured in 6-well plates in a collagen sandwich, according to the suppliers protocol (Invitrogen). In brief, 3 vials (one per individual) were thawed in 50mL CHRM medium (CHRM CM7000, Invitrogen) and centrifuged for 10 minutes at 100g at 4C. After removing the supernatant, the cell pellet was dissolved in CHPM medium (CHPM CM9000, Invitrogen), which contain FBS, at a concentration of 1.0*10^6 cells/mL. Pooled PHH were cultured in collagen-precoated 24-well plates (Gibco) (density: 0.35*10^6 cells/well) and cells were attached to the wells in the incubator for 4 hours at 37C. After the incubation, the debris was removed by shaking and washing the cells twice with Williams medium E (WME CM6000, Invitrogen). Subsequently, cells were covered by 150 L/well of 1.0mg/mL collagen-mixture (containing 10*DMEM, 1.0mg/mL Collagen type 1 (BD), 0.2M NaOH and, MQ) and incubated at 37C for approximately 30 minutes until the collagen was fixed. After that, culture medium (WME + 20mL Hepatocyte Supplement Pack (CM4000, Invitrogen) substituted with 1% penicillin/streptomycin (Gibco)) was added.” Available from: https://www.ebi.ac.uk/biostudies/arrayexpress/studies/E-MTAB-5984

Figure 4 D is missing

Labelling was lost when copied across to journal template. This has been rectified.

line 407 -please check albumin is not a metabolite

“metabolite” was changed to “…and secretion of organic substances” Line 406

Figure 7 - please check font in caption and line 429 - cyto- toxicity

Font has been changed to be consistent with journal template

Discussion

line 486 - 492 Authors cite previous findings, could authors may address or indicate the cells or methods used from the previous findings? 

“…mouse fibroblasts cultured in hydrogels of varying levels of crosslinking and therefore stiffnesswas added to lines 489-490

line 538  - cells from 3D spheroids show a delayed cytotoxic response - did the authors consider to investigate the activities of prominent Cytochrome P450 enzymes?

Transcriptomic analysis of CYPs was conducted on cells immediately liberated from 3D scaffolds – not on hanging drops, and the following passage was added to the discussion to address this comment in relation to feedback from Reviewer 1:

Only one key drug metabolising cytochrome (CYP1A2) was differentially upregulated in 3D HepG2 cells to a ≥2-fold significance level though expression levels were still very low. However, other CYPs with the exception of CYP2S1, CYP24A1 and CYP4F22 were enhanced in 3D cultured hepatocytes but to a level that was still significantly lower than primary cells or human liver. This suggests that 3D culture can improve cell functionality and Phase I metabolism in general, but also highlights how far removed the HepG2 cell line is from native hepatocytes.”  Lines 519-524

The following was also added to make it clear to the reader that transcriptomic analysis was not conducted on spheroids:

To further identify functional differences between 2D/3D primed populations in spheroid structures, transcriptomic analysis could be conducted in the future to determine the pathways involved in the attenuated functional enhancement post-3D priming.” Lines 547-550

Comments on the Quality of English Language

Please check the title and consider to include the "hepatocyte model cell line" used in the study

“Hepatocyte” has been changed to “a model hepatocyte cell line” in the title. Line 3

Line 85 and 94- please check, HepG2 culture or model

“Hepatocyte” was changed to “HepG2” for clarity. Lines 86 and 95

line 260 – 267 text passage redundant

This is the figure legend for the schematic detailed in Figure 1, however due to formatting changes it appears as a separate paragraph. This has been rectified.

line 275 - 277 The authors address the morphological changes - it remaines unclear if the authors refer to differences or primary cells.

“…of HepG2 cells.” was added to the sentence to make it clear the model cell line was being referred to and not primary cells. Line 279

line 154 - please check Hoechst

Spelling rectified. Line 162
